# Photo Activity on Social Networking Sites and Body Dissatisfaction: The Roles of Thin-Ideal Internalization and Body Appreciation

**DOI:** 10.3390/bs12080280

**Published:** 2022-08-11

**Authors:** Changying Duan, Shuailei Lian, Li Yu, Gengfeng Niu, Xiaojun Sun

**Affiliations:** 1Key Laboratory of Adolescent Cyberpsychology and Behavior (CCNU), Ministry of Education, Wuhan 430079, China; 2School of Psychology, Central China Normal University, Wuhan 430079, China; 3College of Education and Sport Sciences, Yangtze University, Jingzhou 434023, China; 4Beijing Normal University Collaboration Innovation Center, Central China Normal University Branch, Wuhan 430079, China

**Keywords:** photo activity on SNS, thin-ideal internalization, body appreciation, body dissatisfaction, sociocultural theory

## Abstract

Objective: According to sociocultural theory, media is associated with detrimental effects on body image. Due to the popularity of social networking sites (SNS) and the prevalence of body image disturbance among young women, the association between them is worth further exploration. This study examined the relationship between photo activity on SNS and body dissatisfaction (BD) and the roles of thin-ideal internalization (TII) and body appreciation (BA) in this relation. Materials and Methods: A total of 746 Chinese female undergraduate students (mean age 20.34 ± 1.47 years) completed a questionnaire measuring SNS photo activity, TII, BD, and BA. Results: (1) Photo activity on SNS was positively associated with BD (*r* = 0.10, *p* < 0.01), and TII could mediate this relation (*β* = 0.07, 95% CI = [0.04, 0.10]). (2) Both the direct effect of SNS photo activity on BD (*β* = −0.08, *p* < 0.05) and the mediating effect of TII (*β* = −0.09, *p* < 0.01) were moderated by BA. Specifically, these associations were more pronounced for students with lower BA. Conclusion: People exposed to ideal photos or images can shape women’s body image perception via TII, whether in the age of traditional media or the Internet, and BA did not buffer the effect of ideal photos on internalization. Our findings could provide practical suggestions for rational photo activity on SNS and the intervention for BD.

## 1. Introduction

In the context of the COVID-19 pandemic lockdown, people were sedentary indoors, and a lack of outdoor sports may have led to weight gain (e.g., belly fat and leg thickness), which can influence individuals’ body image perception. Body image perception is an important dimension of self-concept [1,2], closely related to an individual’s social and emotional adaption. Previous studies found that negative body image perception, mainly BD, is directly associated with maladaptive outcomes, including lower self-worth [3] and even eating disorders [4] or exercise addiction [5]. Emerging adult women usually pay more attention to physical attractiveness than they do in other periods of their life [6,7] and are vulnerable to BD [8], regardless of their socioeconomic status, social culture, and ethnicities [9]. Thus, it is imperative to identify risk and protective factors associated with BD among young women.

BD typically refers to how individuals negatively evaluate their bodies and perceive the discrepancy between their actual and ideal bodies [10]. The factors influencing BD have been widely examined, including family, peers, and media [11,12,13,14]. In particular, appearance-related media are an extremely risky factor for body image [15,16]. In the current information era, a social networking site (SNS) is a popular option throughout the world for people presenting and browsing appearance-related photos, which have a prominent impact on BD [17,18,19]. These sites include options such as Facebook and Instagram in Western countries and WeChat Moments and Qzone in China (accounting for 85.1% and 47.6% of Chinese citizens, respectively) [20]. In the context of the COVID-19 pandemic lockdown, people used SNS to access pandemic-related information [21] or to ameliorate anxiety [22], such as by communicating, working, learning, and seeking entertainment. The average daily time spent on SNS has increased by 2 to 3 h when compared with the pre-pandemic lockdown period [21]. Meta-analyses demonstrated that SNS use (i.e., browsing the photos on SNS) is positively related to body image disturbance [10,23]. However, previous studies mainly focused on the relationship between viewing photos on SNS and BD; studies on other behaviors (i.e., comments and likes) in relation to body image are scarce. Both comments [24] and likes [25] are closely associated with body image self-evaluation. This study aimed to explore the process through which photo activities on SNS (including viewing, commenting, and likes) exercise their influence on BD among young women and identify further individual differences underlying this association, which is of great theoretical and practical significance for the use of SNS, to protect mental health in the context of the COVID-19 pandemic.

### 1.1. SNS Photo Activity and Body Dissatisfaction

Image-based SNS (i.e., Instagram) is an increasingly popular platform where people share ideal photos, solicit feedback (i.e., likes and comments), and browse other people’s photos [26]. Photo activity on SNS refers to viewing, commenting, and liking other people’s photos [27,28], which are common SNS activities. People engage in photo activity on SNS to view socially and physically attractive people and present themselves in that way, to maintain offline relationships, and to get visual quantifiable feedback [29]. However, numerous empirical studies have found that people’s participation in photo activity on SNS, such as browsing idealized photos on SNS [30], photos with positive appearance comments [25], and photos with a large number of “likes” [31], could be deleterious to individuals’ body images or perception (i.e., cause body and facial dissatisfaction). The sociocultural theory [12,13] may shed light on this phenomenon, which points out that people’s appearance dissatisfaction stems from sociocultural body ideals [32]; namely, the unattainability of these appearance ideals may make people feel dissatisfied with their appearance. Photos on SNS are usually appearance-focused images displaying an ideal body image [33,34], such as selfies [35] and fitspiration [36]. These ideal photos are associated with appearance comparison [17,37,38,39], which in turn influence self-evaluation (i.e., BD) [40,41,42]. For instance, experimental studies indicated that female college students who frequently viewed fitspiration images reported more BD than did those who frequently viewed travel images [43,44]. Likewise, experimental research documented that people who viewed manipulated photos of attractive celebrities and peers on SNS tended to display female BD compared with those who viewed travel or original images, regardless of whether the photos have a disclaimer or not [45,46,47,48]. Thus, it was hypothesized that SNS photo activity was negatively associated with BD (Hypothesis 1).

### 1.2. Thin-Ideal Internalization as a Mediator

TII may also be a key process that mediates the association between photo activities on SNS and body dissatisfaction. TII refers to the extent to which individuals subscribe to social standards for physical appearance and aspire to attain these standards [49,50], which iare closely associated with body image. A longitudinal study also suggested that the baseline level of internalization of the media ideal significantly predicted BD at eight months among female adolescents [50]. A meta-analysis indicated that the internalization of body shape ideals is closely positive to BD [51].

TII may also be affected by photo activities on SNS. The cultivation theory provides a perspective to understand the internalization process induced by media, which is that media’s perpetual depiction of certain values, themes, and ideals molds people’s view of social reality [52,53]. Photo activities on SNS, such as selfies, are usually attractive photos selectively modified, edited, and posted by peers or acquaintances around us [33,54] and emphasizing a slim and beautiful appearance [55]. In this situation, repeated exposure to the perfect body or appearance on SNS from peers would make them more likely to assimilate social body standards and perceive those ideals as the goals they should achieve. A meta-analysis documented that appearance-related activity on SNS was a stronger risk factor for internalizing body ideals than the risk of general SNS use among women was [56]. Similarly, many empirical researchers found that appearance-focused photo activities on social media were positively associated with the internalization of social ideals [57,58,59,60]. In addition, under the perspective of the sociocultural theory, the Tripartite Influence model further posits TII as the important mediating mechanism for the effects of the media on body image [12,50]. Under this perspective, empirical studies also verified the mediating role of TII on the effect of appearance-related activities on SNS on body image [28,61]. Thus, it was further hypothesized that TII mediated the relationship between SNS photo activity and BD (Hypothesis 2).

### 1.3. Body Appreciation as a Moderator

Although SNS photo activity may induce BD through TII, not all users may be equally influenced, and the potential individual differences should be examined, which are of great practical significance. BA, an important aspect of positive body image [62,63], is an important indicator for measuring the degree to which individuals respect and love their body image [64]. BA encompasses an accepting, favorable, and respectful attitude toward one’s body while also rejecting media-promoted appearance ideals as the only form of human beauty [65]. It is closely associated with low negative emotion (i.e., depression and anxiety) [66], an absence of eating disorders [67], and high body satisfaction [68]. Moreover, BA is a “protective filtering”, which can buffer the negative effects on body image induced by the thin-ideal images on media [69,70]. For instance, experimental studies found that female college students with higher BA would report fewer appearance discrepancies and BD after exposure to thin-ideal advertisements or ideal media models than those reported by female college students with lower BA [69,71].

Meanwhile, individuals with high BA tend to focus on their positive internal characteristics [72]. Since they are less likely to be affected by unrealistic body standards [65], they are less likely to internalize the sociocultural thin-ideal body, including viewing or commenting on SNS photos. In addition, empirical research also documented that BA could moderate the relationship between the internalization of general attractiveness and adolescent facial dissatisfaction [28]. Namely, BA could ameliorate the deleterious effect of TII on body image. Building on previous research, we further hypothesized that BA moderated the direct and indirect pathways between SNS photo activity and BD (Hypothesis 3).

### 1.4. Our Study

Despite evidence suggesting that photo activity on SNS is a well-established risk factor for women’s body image, the underlying mediation and moderation process remain unclear. Thus, a moderated mediation model was constructed (see Figure 1) to examine the association between the effect of photo activity on SNS on women’s body image and its inner mechanism—the mediating role of TII and the moderating role of BA. This study may shed more light on how photo activity on SNS is associated with BD and individual difference and contribute to understanding the role of photo activity on SNS on Chinese young women’s body image.

## 2. Materials and Methods

### 2.1. Participants

The convenience sampling method was adopted to recruit female students from central China. A total of 780 female undergraduate students voluntarily participated in our study, among which 746 provided valid questionnaires, and the percentage of people who completed the entire survey was 95.64%. All were aged 17–24, with an average age of 20.34 years (SD = 1.47); a total of 145 students were freshmen, accounting for 19.43%; a total of 204 students were sophomores, accounting for 27.35%; a total 215 students were juniors, accounting for 28.82%; a total of 182 students were seniors, accounting for 24.4%.

### 2.2. Procedure

The study acquired approval from the ethics committee to recruit participants from Central China Normal University. The survey tool took place between 5 March and 30 March 2022. All participants were recruited via online advertisements to complete a paper questionnaire voluntarily. Before collecting the data, the written informed consent of each participant was obtained. A well-trained graduate student then administered a pencil-and-paper survey in classrooms, and all data were collected anonymously in Mandarin. Within 15–20 min, participants completed a survey including demographic variables, social networking sites’ photo activity, TII, BD, and BA. After the assessment, all participants were offered a small gift as an incentive.

### 2.3. Measurement

#### 2.3.1. Social Networking Sites’ Photoactivity

The Instagram Selfie-Viewing index [27] was adopted to assess participants’ photo activities on SNS. We changed Instagram to Chinese social networking sites such as WeChat Moments or Qzone. Participants were asked to respond to the five items on a 5-point scale from 1 (never) to 5 (always), with higher scores indicating a higher tendency to browse, comment, and like photos on SNS. A representative item was “Browse photos or videos of friends on WeChat Moments or Qzone”. This measure translated to Chinese had good reliability and validity. The confirmatory factor analysis (CFA) indicated a good construct validity—χ^2^/df = 4.64. RMSEA = 0.07, CFI = 0.99, and TLI = 0.96. Furthermore, Cronbach’s alpha was 0.79 in our study.

#### 2.3.2. Body Appreciation

The Chinese version of the BA Scale (BAS-2) [73] assessed BA. Participants rated ten items on a 5-point scale from 1 (never) to 5 (always), with high scores indicating greater BA. A representative item was “I feel good about my body”. The scale had good reliability (α = 0.92) in the Chinese sample (66). In our study, Cronbach’s alpha was 0.90.

#### 2.3.3. Body Image Dissatisfaction

The Chinese version of the Body Image Depression scale [74] was used. It included 25 items (e.g., I think my weight is overweight), and participants were asked to respond on a 3-point scale from 1 (agree) to 3 (not agree), with a higher score indicating a higher tendency to feel dissatisfied with their body. The scale has been used in Chinese university students with good reliability and validity [75]. In our study, Cronbach’s alpha was 0.87.

#### 2.3.4. Thin-Ideal Internalization

TII was assessed by the Internalization: Thin/Low Body Fat subscale of the Sociocultural Attitudes Toward Appearance Questionnaire-4-Revised (SATAQ-4R) [76]. Participants rated five items on a 5-point scale from 1 (definitely disagree) to 5 (definitely agree), with high scores indicating people have a greater level of thin-ideal internalization. A representative item was “I want my body to look very thin”. This measure was translated to Chinese and found to have good reliability and validity. The confirmatory factor analysis (CFA) indicated a good construct validity—χ^2^/df = 2.98. RMSEA = 0.05, CFI = 0.99, and TLI = 0.97. In our study, Cronbach’s alpha was 0.70.

## 3. Results

### 3.1. Descriptive and Correlational Analysis

Pearson’s correlation analyses in SPSS were used to test the correlations between the main variables. As shown in Table 1, SNS photo activity was positively associated with TII, BD, and BA. TII was positively associated with BD and negatively associated with BA. BA was negatively associated with BD.

### 3.2. Testing for Mediation Model

The PROCESS macro (Model 4) in SPSS [77] was used to test whether TII mediated the link between SNS photo activity and BD (Hypothesis 2). Regression analysis revealed that: (1) SNS photo activity positively predicted BD (*β* = 0.30 and *p* < 0.001); (2) when TII entered the regression analysis, SNS photo activity also positively predicted BD (*β* = 0.23 and *p* < 0.001) and TII (*β* = 0.22 and *p* < 0.001), and TII positively predicted BD (*β* = 0.31 and *p* < 0.001). The indirect effect of SNS photo activity on BD via TII was significant; to be clear, ab = 0.07, *SE* = 0.01, and 95% CI = [0.04, 0.10]. The indirect effect accounted for 23.36% of the total effect. The mediation analysis indicated that TII partially mediated the relationship between SNS photo activity and BD (see Table 2). Therefore, Hypothesis 2 was supported.

### 3.3. Testing for Moderated Mediation

The PROCESS macro (Model 59) in SPSS [77] was further used to test whether BA would moderate the direct and indirect relationships between selfie-viewing and BD via appearance comparison. Specifically, we estimated the moderating effect of BA on relations between (1) SNS photo activity and BD; (2) the first mediation path of SNS photo activity and TII; (3) the second mediation path of TII and BD.

As Table 3 illustrates, the interaction between SNS photo activity and BA on TII was insignificant (*β* = −0.02 and *p* > 0.05), suggesting the relationship between SNS photo activity and TII was not moderated by BA. However, the interaction between SNS photo activity and BA on BD was significant (*β* = −0.08, *p* < 0.05), indicating that the relationship between SNS photo activity and BD was moderated by BA. We used simple slope tests to analyze the relationship between SNS photo activity and BD for higher (+1 SD) and lower (−1 SD) BA (see Figure 2). Although both slopes were significant, the relationship between SNS photo activity and BD was weaker at higher BA (*β*_simple_ = 0.18, SE = 0.04, and 95% CI = [0.09, 0.26]) than it was at lower BA (*β*_simple_ = 0.34, SE = 0.05, and 95% CI = [0.24, 0.43]).

Additionally, the interaction between BA and TII on BD was significant (*β* = −0.09 and *p* < 0.01). We used simple slope tests to analyze the relationship between TII and BD for higher (+1 SD) and lower (−1 SD) BA. As shown in Figure 3, the two slopes were significant, and the relationship between TII and BD was weaker at higher BA (*β*_simple_ = 0.19, SE = 0.05, and 95% CI = [0.10, 0.29]) than it was at lower BA (*β*_simple_ = 0.37, SE = 0.04, and 95% CI = [0.28, 0.45]). Thus, Hypothesis 3 was partially supported.

## 4. Discussion

Based on current research and the real-life experiences of young women, our study examined the association between photo activity on SNS and female college students’ BD and its underlying mechanism. The results show that photo activity on SNS is indirectly associated with BD through the mediating effect of TII. In addition, BA moderates the direct effect of photo activity on SNS on BD and the second link of the indirect effect (i.e., TII on BD). These findings advance our understanding of how and when photo activity on SNS is related to body image via TII and BA, which benefits the prevention and intervention of BD.

First, as hypothesized, the results suggest that photos on SNS are positively associated with BD, which is in accord with the main points of the sociocultural theory [12,13]. Previous studies suggested that media pressure is a risk factor for women’s body image since it promotes an unrealistic thin body and appearance ideal [78,79]. SNS is acquaintance-centric, and most of the photos on SNS are posted by our peers or friends who are presenting an idealized version of themselves. Especially in the COVID-19 pandemic lockdown, people stayed indoors and had more time to manipulate these photos in terms of beauty or thinness [80]. Meanwhile, these photos reflected public norms and perceptions of beauty and represented contemporary popular beauty ideals [33,81,82], which likely drew young women to pay attention to their appearance and body image [33,83]. In addition, emerging adult women pay more attention to physical attractiveness than they do in other periods of their life [6,7], and they are more likely to refer to these idealized images, compare themselves with idealized content [59,84] because of peer similarity, and further develop negative perceptions of body image. At the same time, in the context of the COVID-19 pandemic lockdown, people staying indoors and lacking exercise made it difficult for most women to achieve the idealized standards, which increased the risk of BD. We should pay attention to the negative influence of the ideal body image portrayed by social media on women’s body perception.

Second, as hypothesized, the results suggest that TII mediated the association between photo activity on SNS and BD. The findings of this study are consistent with the Tripartite Influence model [12,50]; namely, social media is negatively associated with body image through TII. We found that people engaging in activities on SNS were more likely to internalize their ideal appearance, which in turn decreased their body satisfaction. As we all know, SNS photos are usually carefully edited and selected by our peers to be “perfect”, appearance-focused photos, presenting the best physical characteristics [85]. Meanwhile, the number of likes and comments a photo receives are a symbolic form of validation from others and reflect our norm and perception of social ideal beauty appearance [24,33,86,87]. As the cultivation theory emphasizes, social reinforcement and peer influence may promote the cognitive internalization of a standard for physical attractiveness [53,88], which leads to women monitoring their bodies, comparing them with beauty ideals [50,89], and inducing negative BD. Specifically, people are more likely to unintentionally and automatically internalize thin-ideal images when they browse a thin and attractive photo and concurrently compare their bodies with the ideal body [90]. Dissatisfaction with one’s body occurs when people accept these ideal standards and fail to meet them. These findings verify the Tripartite Influence model and indicate that SNS might be associated with body image as much as traditional media is, which advances our understanding of the role of media on body image.

In addition, our hypothesis was partially supported; BA moderated the direct effect of photos activity on SNS and BD and the second pathway of the mediating effect of TII, while it did not moderate the first pathway of the mediating effect of TII.

Contrary to our hypothesis, our results found that BA did not moderate the association between photos activity on SNS and TII. According to cultivation theory, the repeated exposure to overwhelming thin-ideal information on media may influence our perception of the body image that “thinner is the better figure”; subsequently, they deeply assimilate thin-ideal into their perception of body image [53,91]. At the same time, with the popularity of the notion “thinner is better”, our higher BA for our bodies may also stem from lower body mass index [92]. As such, it is difficult to change an individual’s perception of beauty; BA cannot help women to resist the media-induced internalization of the thin-ideal body type.

Although BA did not mitigate the effect of photo activities on TII induced by media, it could reduce the likelihood of BD induced by browsing ideal photos and TII. To be specific, BA moderated the direct effect of photos on SNS and BD and the second pathway of the mediating effect of TII, with these negative associations being more noteworthy for female undergraduate students with lower BA. These findings are consistent with those shown by Wang et al. [28] and Andrew et al. [69]. They suggested that BA could act as a protective factor in attenuating the relationship between “perfect” photo culture and women’s body image, as well as the association between TII and BA. Specifically, individuals with high BA have more positive coping strategies for appearance-related information on media than people with low BA. Since individuals with high BA possess body-related information more positively, with more positive feelings (i.e., respect and love) about their body, they usually invest less in their self-worth in their body or appearance [65,72] than do those with more negative feelings about their body, and they pay less attention to the weight-related information [93] emphasized in the media. Therefore, people with high BA report less BD, even if they view or “like” ideal photos on SNS and internalize the thin-ideal.

## 5. Implications and Limitations

Several limitations should be noted. First, a cross-sectional design was used. Future studies should adopt a more ecologically valid format (i.e., browse SNS images on an iPad or phone) and experimental design to strengthen the causal relationship. Second, participants’ body mass index was not collected in the present study, as previous have shown that both normal-weight women and overweight women expressed BD and desired a thinner body [94]. Future research could adopt it as a control variable. Third, we used a self-reported scale of SNS photo activities, including viewing, liking, and commenting on photo/video behaviors. Future studies could explore how specific photo activities (viewing, likes, and commenting behaviors) impact body image. Simultaneously, our study only used a nationally representative sample of Chinese female undergraduates, and there are cultural differences in perception of beauty and BA [95], limiting the generalizability of findings to participants in other cultures; future studies should compare cultural differences in photo activities on SNS and body image.

Despite these limitations, testing both the mediating processes and individual differences is essential for a theoretical understanding and practical intervention on the impact of media on body image. From the theoretical perspective, firstly, this study enriched the research for the sociocultural theory and Tripartite Influence model in the area of the Eastern culture’s background, and the results are in accordance with the findings of other cultures (the United States and Australia), contributing to our understanding of the process involved with the photo activities on SNS influencing body image. Secondly, our results deepen the understanding of the protective role of BA [96] in young women’s body image by shedding light on how and when photo activity on SNS is associated with BD. From the practical perspective, based on the main findings, social health curricula and educated programs could be designed to improve BA among young women, specifically by helping young women to evaluate all aspects reasonably and encouraging them to accept and respect their physical appearance. In addition, during the COVID-19 pandemic lockdown, people could have reduced their use of social media (e.g., WeChat Moments and Qzone) to avoid the effect of various harmful information (e.g., appearance or disease) on mental health, and they should be aware that most photos on SNS are idealized, edited, and unrealistic in order to decrease the possibility of internalization and BD.

## Figures and Tables

**Figure 1 behavsci-12-00280-f001:**
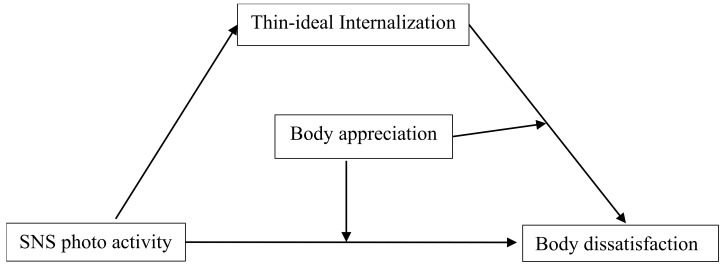
The proposed moderated mediation model.

**Figure 2 behavsci-12-00280-f002:**
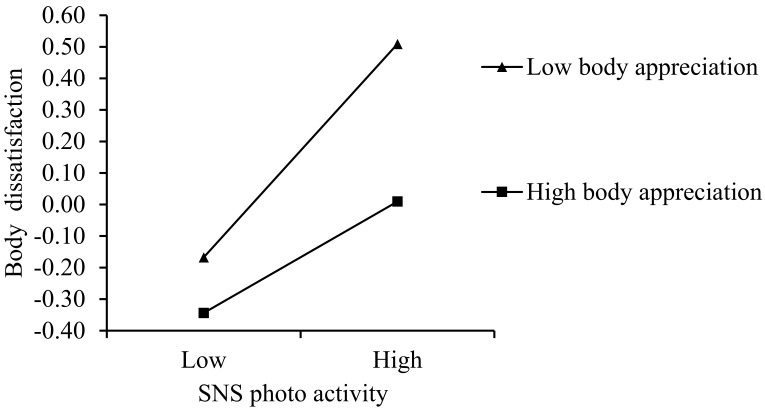
Body appreciation moderated the relationship between SNS photo dissatisfaction and body dissatisfaction.

**Figure 3 behavsci-12-00280-f003:**
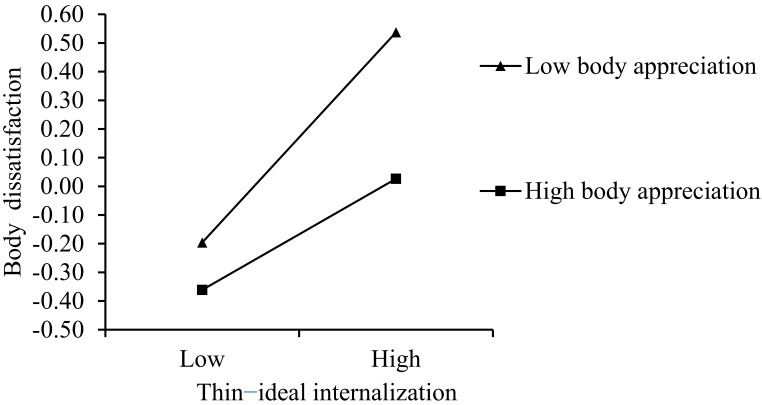
Body appreciation moderated the relationship between thin-ideal internalization and body dissatisfaction.

**Table 1 behavsci-12-00280-t001:** Descriptive statistics and correlation matrix between variables.

Variables	*M* (*SD*)	1	2	3	4
1. SNS photo activity	3.32 (0.76)	1			
2. Thin-ideal internalization	3.37 (0.78)	0.22 **	1		
3. Body dissatisfaction	1.91 (0.46)	0.30 **	0.36 **	1	
4. Body appreciation	4.03 (0.60)	0.10 **	−0.07 *	−0.18 **	1

Note: * *p* < 0.05; ** *p* < 0.01; SNS photo activity = social networking sites’ photo activity.

**Table 2 behavsci-12-00280-t002:** Testing the mediation effect of selfie-viewing on body dissatisfaction.

Predictors	Model 1 (Body Dissatisfaction)		Model 2 (Thin-Ideal Internalization)		Model 3 (Body Dissatisfaction)	
	*β*	*t*	*β*	*t*	*β*	*t*
Age	0.02	0.56	0.05	1.36	0.01	0.14
SNS photo activity	0.30	8.47 ***	0.22	6.27 ***	0.23	6.66 ***
Thin-ideal internalization					0.31	9.05 ***
*R* ^2^	0.09		0.05		0.18	
*F*	35.86 ***		20.19 ***		53.83 ***	

Note: *** *p* < 0.001; The research variables (excluding demographic variables) in the regression model were standardized.

**Table 3 behavsci-12-00280-t003:** Testing the moderated mediation effect of selfie-viewing on body dissatisfaction.

Predictors	Model 1 (Thin-Ideal Internalization)		Model 2 (Thin-Ideal Internalization)	
	*β*	*t*	*β*	*t*
Age	0.05	1.40	−0.01	0.14
SNS photo activity	0.24	6.56 ***	0.26	7.69 ***
Thin-ideal internalization			0.28	8.38 ***
Body appreciation	−0.10	−2.72 **	−0.17	−5.11 ***
Interaction 1	−0.02	−0.61	−0.08	−2.51 *
Interaction 2			−0.09	−2.69 **
*R^2^*	0.06		0.23	
*F*	12.08 ***		36.99 ***	

Note: * *p* < 0.05; ** *p* < 0.01; *** *p* < 0.001; Interaction 1 = SNS photo activity * Body appreciation; Interaction 2 = Thin-ideal internalization * Body appreciation.

## Data Availability

The data of this study are available from the corresponding author upon reasonable request.

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
