# Peer review of "Photo Activity on Social Networking Sites and Body Dissatisfaction: The Roles of Thin-Ideal Internalization and Body Appreciation"

_behavsci, 2022, doi:10.3390/bs12080280_

Round 1

Reviewer 1 Report

According to the authors, the objective of this study was not only to examine the relationship between photo activity on SNS and body dissatisfaction in Chinese female undergraduate students but also to elucidate the roles of thin-ideal internalization and body appreciation in this relation. In a general way, the study was well conducted, mainly by the questionnaires used (Chinese adaptation of Instagram Selfie-Viewing index, Chinese version of the Body Appreciation Scale, Chinese version of the Body Image Depression scale and Sociocultural attitudes towards appearance questionnaire-4-revises) and also by the statistical analysis, and it presented interesting data. However, there are few points that should be concerned.

In the Abstract section:

1) Please, provide the average age and standard deviation of the sample.

2) Please, add the statistical significance found in the study for relevant results described in this section.

3) In the abstract authors describe that “students completed a questionnaire measuring SNS photo activity, appearance comparison, body dissatisfaction, and body appreciation”. Actually, appearance comparison is not measured in the study. It is better to change “appearance comparison” with “Thin-ideal internalization”.

In the Introduction section:

4) On page 2, lines 61 to 63, the authors cited that " However, previous studies mainly focused on the relationship between viewing photos on SNS and body dissatisfaction, other behaviors (i.e., comments and likes) in relation to body image are scarce”. Please, cite references.

5) Considering that in the study a lot of emphasis is placed on SNS photo activity, it would be helpful to include a brief description of image-based SNSs, such as Instagram.

In the Material and Methods section:

6) In the introduction authors describe the context of the COVID-19 pandemic lockdown. In the procedure is not clear if the study is conducted in pandemic period. Please specify the period in which the study was conducted.

7) Please, add the number of Ethical approvals.

8) In the subsection "Descriptive and Correlational Analysis", the authors mentioned the associations between variables. Please, add more information to clarify what tests were used.

In the Results section

9) On page 5, lines 215 to 216, the authors cited that "Body appreciation was negatively associated with body dissatisfaction”. Is better to use the variable name used throughout the text, i.e. Appearance appreciation.

10) In results section, authors describe Hypothesis 3a and 3b, but they are not mentioned before.  Please include these Hypothesis in the introduction.

In the Discussion section:

11) On page 8, lines 286 to 288, the authors cited that "Especially in the 287 COVID-19 pandemic lockdown, people stayed indoors and had more time to manipulate these photos into beauty or thinness.”. Please, cite references.

12) On page 8, line 291 “[32,81].In addition, emerging adult women pay more attention to physical attractiveness”, please correct the spacing errors.

Reviewer 2 Report

Good work. No relevant errors was appreciated.
